# Effectiveness of Digital Physiotherapy Practice Compared to Usual Care in Long COVID Patients: A Systematic Review

**DOI:** 10.3390/healthcare11131970

**Published:** 2023-07-07

**Authors:** María-José Estebanez-Pérez, Rocío Martín-Valero, Maria Jesus Vinolo-Gil, José-Manuel Pastora-Bernal

**Affiliations:** 1Department of Physiotherapy, Faculty of Health Science, University of Malaga, 29071 Málaga, Spain; rovalemas@uma.es (R.M.-V.); jmpastora@uma.es (J.-M.P.-B.); 2Department of Nursing and Physiotherapy, Faculty of Nursing and Physiotherapy, University of Cadiz, 11009 Cadiz, Spain; mariajesus.vinolo@uca.es

**Keywords:** digital physiotherapy practice, telerehabilitation, telemedicine, Long COVID, persistent COVID syndrome

## Abstract

Long COVID syndrome has been recognized as a public health problem. Digital physiotherapy practice is an alternative that can better meet the needs of patients. The aim of this review was to synthesize the evidence of digital physiotherapy practice in Long COVID patients. A systematic review was carried out until December 2022. The review was complemented by an assessment of the risk of bias and methodological quality. A narrative synthesis of results was conducted, including subgroup analyses by intervention and clinical outcomes. Six articles, including 540 participants, were selected. Five articles were considered of high enough methodological quality. Parallel-group, single-blind, randomized controlled trials were the most commonly used research design. Tele-supervised home-based exercise training was the most commonly used intervention. Great heterogeneity in clinical outcomes and measurement tools was found. A subgroup analysis showed that digital physiotherapy is effective in improving clinical outcomes. Significant differences in favor of digital interventions over usual care were reported. Nevertheless, discrepancies regarding effectiveness were found. Improvements in clinical outcomes with digital physiotherapy were found to be at least non-inferior to usual care. This review provides new evidence that digital physiotherapy practice is an appropriate intervention for Long COVID patients, despite the inherent limitations of the review. Registration: CRD42022379004.

## 1. Introduction

The syndrome known as Long COVID or persistent COVID has been defined by the World Health Organization (WHO) as a condition that occurs in people with probable or confirmed SARS-CoV-2 infection with symptoms lasting at least 2 months and that cannot be explained by an alternative diagnosis [1,2]. Symptoms are wide-ranging and fluctuating [3,4] and can include fatigue and shortness of breath dysfunction. Over 200 different reported symptoms present against daily functions, job position, health perception, and mood, among others [5,6,7,8]. A very similar definition was provided by The National Health Service in England (NHS), who defined Long COVID as the signs and symptoms that develop during or after COVID-19 and continue for more than 12 weeks and are not explained by an alternative diagnosis [9]. To add more information, we include the definition of The National Research Action Plan on Long COVID and the Services and Supports for the Longer-term Impacts of COVID-19 from the United States government, which defined Long COVID as signs, symptoms, and conditions that continue or develop after initial COVID-19 or SARS-CoV-2 infection. The signs, symptoms, and conditions are present four weeks or more after the initial phase of infection; may be multisystemic; and may present with a relapsing–remitting pattern and progression or worsening over time, with the possibility of severe and life-threatening events even months or years after infection. It represents many potentially overlapping entities, likely with different biological causes and different sets of risk factors and outcomes [10].

Long COVID has been recognized as a public health problem; therefore, interventions that support patient management are critical to reducing the disease burden [11]. Symptoms persisted for more than six months with at least one sequela, requiring ongoing rehabilitation and evaluation, as has been stated [12], with a significant impact on reduced quality of life, capacity to work, and performance of usual daily activities [13]. The long-term effects of the disease are not related to the severity of the initial infection; they can affect young and adult fit patients and those who did not visit the hospital with COVID symptoms [9].

A clinical guideline for Long COVID patients was developed, including the recommendation of physiotherapy interventions and strongly advocating a multidisciplinary rehabilitation approach [14]. Digital physiotherapy practices, are methods and protocols for carrying out the rehabilitation process remotely, with or without supervision, and may also be referred to as telehealth, telemedicine or telerehabilitation [15]. The digital physiotherapy practice has been suggested as an innovative strategy in the management of COVID-19 disease [16] and its sequelae [17], aiming to increase accessibility and improve continuity of care [18]. Recent research showed that digital physiotherapy interventions could improve functional capacity and exercise perception and could be applied with minimal adverse impacts [19]. Traditional rehabilitation interventions seemed to improve muscle strength, dyspnea, walking capacity, functional capacity and quality of life; nevertheless, results on pulmonary function were inconsistent [20]. Notwithstanding, a recent meta-analysis states that telerehabilitation may be an effective and safe solution for survivors of COVID-19 [21]. However, to date, the evidence of digital interventions is limited, with low certainty of evidence [22], and systematic reviews have pointed out that clinical and economic effectiveness are still lacking [23].

In view of the increasing publication of randomized controlled trials that have not been reviewed to date, a thorough and rigorous review of methodological quality is recommended. On the basis of the diverse nature of digital physiotherapy interventions [24], more research is needed to improve our understanding of why particular interventions are or are not successful [25].

As such, the primary aim of this systematic review was to explore the effectiveness of digital physiotherapy practice interventions compared to usual care for adult patients with Long COVID by reporting the main changes in outcomes. Secondary aims were to describe the characteristics of the digital interventions.

## 2. Materials and Methods

### 2.1. Identification Data Sources and Search Strategy

This systematic review was performed following the Preferred Reporting Items for Systematic Reviews and Meta-Analyses (PRISMA) [26] statement, as well as a synthesis of the findings of all evidence published following the methodological recommendations of the Cochrane Collaboration Handbook. The PRISMA checklist is detailed in Appendix A, Figure A1.

A systematic search in the following databases was conducted: PubMed/MEDLINE, Cochrane Library, PeDro (Physiotherapy Evidence Database), Embase, CINAHL database, Scopus, EBSCO, Prospero, Google Scholar, Tryp database, and NICE. Search was conducted in Title, Abstract, and Keywords. The search strategy combines terms included in MeSH related to the population and intervention used: “Long COVID” OR “persistent COVID syndrome” AND “Digital Physiotherapy Practice” OR “Telemedicine” OR “Telerehabilitation”. The search strategy with keywords is reported in Appendix A, Table A1.

Search includes publication dates from 2019 to 2022 in any language. For trials published in lesser-used languages, a translated version was sourced. Abstracts and articles were screened for further eligibility. In addition, a search was performed on the ClinicalTrials.gov registry website in order to locate ongoing and unpublished trials. A hand search of references was also performed for further relevant bibliographies.

### 2.2. Study Selection

In line with the PRISMA guidelines, the inclusion and exclusion criteria were established through the definition of the PICO (population, intervention, comparison, and assessment) strategy. The PICO acronym (Patient/Population–Intervention–Comparison/Comparator–Outcome) [27] was used with the intention to answer the research question: Is the practice of digital physiotherapy effective to improve clinical outcomes in patients with Long COVID compared to usual care?

Patients:

Adults [≥18 years] with a diagnosis of Long COVID syndrome. ICD-10 (U09) e ICD-11 (RA02) [28]. Since the beginning of the pandemic, the classification and terminologies have been progressively activating emergency codes for COVID-19 in ICD-10 and ICD-11. A set of additional codes were activated to document flag conditions that occur in the context of COVID-19 [28].

Intervention:

Any treatment intervention, synchronous or asynchronous, provided via digital physiotherapy practice or rehabilitation services at a distance. The intervention must have been a practice in any area of physical therapy, as defined by the World Confederation for Physical Therapy [29], remotely or outside of a regular session by a physiotherapist thanks to new technologies.

Comparison:

Digital physiotherapy practice compared with usual face-to-face rehabilitation treatments, center-based rehabilitation treatments, or usual care and educational care for Long COVID symptoms.

Outcomes:

As primary outcomes, any clinical outcome measure (pulmonary capacity, dyspnea, daily life activities, functional capacity, health-related quality of life, muscle strength, balance, cardiovascular parameters). Secondary outcomes may include satisfaction with care, participant experience, adherence, and adverse effects.

Study Design:

Only Randomized clinical trials [RCTs] were included.

The exclusion criteria were:

Telehealth interventions for monitoring symptoms or physiological parameters only (i.e., telemonitoring). Studies where the comparison group received no usual care, no treatment, or no rehabilitation (waiting list) will be excluded.

### 2.3. Data Extraction

Titles and abstracts were screened by two reviewers. If an article appeared to be potentially relevant, it was retrieved as a full-text article and assessed to see if it fulfilled the criteria for inclusion/exclusion. If a consensus could not be reached, a third or fourth reviewer was consulted. The reviewers identified and excluded duplicates. Following the full-text analysis, a decision was made as to which articles must be included in the final review. Study characteristics and outcomes data were collected, including eligibility criteria, sample size, age and country of recruitment, type of intervention, session frequency, program duration, delivery format, outcomes measures, assessment time points and follow-up.

### 2.4. Evaluation of Methodological Quality and Risk of Bias

PeDro scale, based on the Delphi list, was used to evaluate methodological quality and risk of bias was evaluated using the Cochrane risk-of-bias tool for randomized trials (RoB 2) [30]. The PeDro scale is made up of 11 criteria that assess internal validity. The PeDro scale scores 10 items (the eligibility criteria do not contribute to the total score). Articles are rated present (1) or absent (0), and each trial is given a total PeDro score ranging from 0 to 10 [31]. It will be considered a low-risk study with high methodological quality with scores equal to or greater than 5 [32].

Assessment of the risk of bias in individual studies was performed as recommended by the Cochrane Collaboration Handbook. Version 2 of the Cochrane risk-of-bias tool for randomized trials (RoB 2) is a recommended instrument to assess the risk of bias in randomized trials included in a systematic review [33]. RoB 2 is structured into a fixed set of bias domains, focusing on different aspects of trial design and reflecting current understanding of how causes of bias may influence study results and the most appropriate ways to assess this risk. Each domain was classified as “low risk of bias”, “some concerns” or “high risk of bias” [34,35].

### 2.5. Data Synthesis and Analysis

The results of the included studies were analyzed through separate narrative syntheses. The data was organized in an Excel spreadsheet and was described as follows: authors/year, design study, risk of bias, intervention characteristics and duration, outcome measures, follow-up and results. Results include all available data.

## 3. Results

The review was conducted according to the registered protocol: CRD42022379004. Figure 1 presents this study’s selection process in a Flowchart, as recommended in the PRISMA statement [36], which shows the total number of retrieved references and the number of included and excluded studies.

The electronic search strategy identified a total of 1693 records from the selected databases. After screening titles, abstracts and reference lists, 353 potentially relevant records underwent full-text review. Of these, six randomized controlled trials were included [37,38,39,40,41,42]. Full-text articles excluded did not meet the eligibility criteria and were excluded due to an ineligible study design, population, intervention, or comparator. Data was extracted from all the study’s reports wherever possible.

### 3.1. Risk of Bias and Methodological Quality

Table 1 shows an evaluation of the methodological quality using the PeDro scale. Studies included in the review had scores of two to nine. High enough methodological quality was considered if they had a score of at least five [43]. We found one study with a PeDro score of nine [40], which is considered “excellent”, three studies with scores between six and eight [37,38,42], which are considered “good”, one study with a score of five [39], considered “fair”, and one study with a score < six, considered ‘poor’ [41].

Figure 2 shows a risk of bias summary using the Cochrane Risk of Bias. A risk of bias graph is shown in Figure 3. A low risk of bias was found in four studies [37,38,40,42], some concerns of bias were found in one [39], and a high risk of bias was found in one study [41].

### 3.2. Characteristics of the Included Trials

Table 2 shows the synthesized findings. Six RCTs with a total of 540 participants met the inclusion criteria and were considered. All studies included patients with Long COVID Syndrome aged from 18 to 75 years. All participants were assigned from their home hospital and complied with ICD-10 Diagnosis. The results of this review cover a wide geographical diversity of participants from China [37,42], the United Kingdom [38], Brazil [39], Spain [40] and India [41].

The results of the articles included in the review show great heterogeneity in terms of interventions, effect sizes reported, clinical outcomes and instruments used. As examples, the studies that evaluate lung capacity reveal differences in the intervention approach as well as in the measurement instruments used. Heterogeneity occurs when there are methodological discordances among the trials included in the review: when the patient populations and the disease or symptoms are not exactly the same in their characteristics; when the outcome variables used are not defined exactly the same, nor are they measured in exactly the same way; when the interventions applied to the patients are not exactly the same, although they bear the same name; and when some of the included trials present some bias in their results, as has been previously stated [44].

Regarding the sample size, four of the articles [37,38,40,42] included a population of more than eighty subjects. In two of the studies [39,41] intervention groups did not exceed fifteen subjects. Parallel-group, single-blind, randomized controlled trials were the most commonly used research designs. Only one study included four arms [40].

#### 3.2.1. Interventions

Regarding interventions, tele-supervised home-based exercise training was the most commonly used method in four studies [37,38,39,40], including 233 participants. A group-guided session was found with semi-structured participant discussion [38]. Two studies opted for an unsupervised program with one weekly teleconsultation, including a total of 74 participants [41,42]. Control groups received short educational instructions at baseline [37,42] or usual care [38,39,41] by continuing their clinical management and any other clinical services, or online supervised respiratory muscle training [40]. A subgroup analysis by intervention is presented in Table 3.

#### 3.2.2. Outcomes

With reference to clinical outcomes and measurement, we found great heterogeneity in pulmonary capacity, quality of life, dyspnea, functional capacity, cognitive and psychological status, exercise tolerance, fatigue, cardiovascular function, and participants’ experiences. A subgroup analysis by outcomes is presented in Table 4.

#### 3.2.3. Results of Articles

A subgroup analysis of the results from the digital physiotherapy intervention showed that five studies stated positive effects on pulmonary capacity [37,38,39,40,42], but only three studies showed significant differences compared to the control group [38,39,40]. Improvements in quality of life were found [37,38,40,42], and mental components in the two articles improved without significant differences from the control group [37,42]. Digital physiotherapy practice has also shown improvement in dyspnea in four articles [37,38,41,42]. In one of the studies, the results of the intervention showed significant differences compared to the control group [41]. In studies without significant differences in improvements [37,38,42], only short-term effects were found. Improvements in functional capacity were found in four studies [37,39,40,42], and significant differences with the control group were found in three of them [37,40,42].

From a cognitive and psychological standpoint [38,40], improvements were observed in the intervention group, although differences were not statistically significant.

Fatigue improvement was reported in one of the studies, with significant results compared to the control group [41]. Exercise tolerance was superior over the control group, but between time and group factors, there were no statistically significant interactions [40]. Cardiovascular function improvements due to digital physiotherapy intervention were found in one study with significant differences from the control group [39]. And finally, participants’ positive experiences [38] were reported, suggesting that participants had improvements that were meaningful to them and even small improvements in measured Quality of life. The level of significance was set at *p* < 0.05. A subgroup analysis of the results is shown in Table 5.

## 4. Discussion

Our systematic review was carried out to analyze the effectiveness of digital physiotherapy practice to improve clinical outcomes in patients with Long COVID compared to usual care. The results of our review are in line with those shown in previous investigations, adding improvements in pulmonary capacity and function. The authors want to highlight the existence of significant improvements versus control groups in three studies, which provides important new knowledge for Long COVID patients.

Regarding the results of rehabilitation interventions, a previous meta-analysis [20] synthesized the effects in COVID-19 patients, concluding that standard rehabilitation seemed to improve dyspnea, anxiety, kinesiophobia, muscle strength, walking capacity, sit-to-stand performance, and quality of life; however, results on pulmonary function were inconsistent. Another systematic review in 2022 focused on telerehabilitation interventions [45], where COVID-19 and post-COVID-19 patients were included and pooled, and concluded that telerehabilitation effects on pulmonary function remain very uncertain with very low certainty of evidence.

Digital physiotherapy practice and synonymous terms have been positioned as a viable alternative intervention for COVID-19 patients and their sequelae; nevertheless, the authors feel it is necessary to discuss some key points identified during this review.

Firstly, this systematic review is focused on Long COVID patients based on the WHO definition of inclusion criteria. This allows us to align with the most current definitions of persistent symptoms and to minimize heterogeneity in the participants of the included studies, which represents an important differentiation with respect to previous reviews [19,20,45].

Secondly, our review includes as an inclusion criterion the existence of control groups receiving usual face-to-face rehabilitation treatments, center-based rehabilitation treatments, or usual care and educational care at least, avoiding statements of the effects of digital physiotherapy practice versus non-interventions that have been included in previous reviews [38]. Furthermore, reviews that combined face-to-face, home exercise programs, and digital interventions as experimental groups imply some limitations of the evidence available for each intervention. In our review, only physiotherapy digital interventions in any area defined by the World Confederation for Physical Therapy were included.

With regard to methodological quality, internal validity and the risk of bias have been assessed through validated tools. Evidence-based practice encourages the integration of high-quality evidence into clinical decision-making for patient care. In turn, low-quality clinical trials can lead to misinterpretations in the systematic reviews that combine them [5,46]. Nevertheless, the authors wish to emphasize that the inclusion of low-quality studies in this systematic review is essential to avoid bias and noted that one of the included studies [41] has a high risk of bias and low internal validity, so results should be interpreted as such, avoiding selection and interpretation biases.

Numerous feasibility studies and clinical trials are currently ongoing, including preliminary results in Long COVID patients [47], which require a continuous review of updated knowledge. Research priorities from WHO and the Long COVID Forum Group with respect to Long COVID involve improving clinical characterization and research and development of therapies [48]. Clinical characterization of patients with Long COVID is essential to providing appropriate treatment options [49]. In this sense, the inclusion criteria regarding digital physiotherapy interventions and clinical outcomes in our review allow us to contribute new, specific knowledge to the scientific demands.

With regard to quality of life, this review shows improvements due to digital interventions; however, in some studies, the improvements compared to the control group are minimal or nonexistent in physical or mental components, which can be interpreted in line with the results published in previous reviews. Some authors have noted that, interestingly, the affective component has improved substantially, suggesting the emotional impact it had [38].

Another aspect we would like to point out is the use of qualitative criteria. In a qualitative interview study in the United Kingdom, online peer support helped patients overcome feelings of inadequate care from healthcare professionals. Furthermore, patients reported feeling less alone and more validated after using online peer support [50]. According to studies indicating that the role of social support for Long COVID patients seems relevant, given that many of them do not feel that they are treated or cared for seriously [51]. Tele-supervised interventions allow more contact with the patient, which may be a clear explanation, as has been previously stated [52].

With regard to the variety and modalities of digital interventions, and despite the variety and quantity of available alternative models implemented globally [53], this review shows scarce intervention alternatives. Tele-supervised home-based exercise training was the most commonly used method. No other digital physiotherapy practices, such as artificial intelligence, virtual reality or video games, currently in common use met the inclusion criteria for this review [54,55,56].

Our review confirms improvements and significant differences from usual care when digital physiotherapy practice is used in Dyspnea, Functional capacity, Cognitive and psychological status, Fatigue, Exercise Tolerance, Cardiovascular function, and Participants’ experiences. These results are in line with the results published in previous reviews on different COVID patients and bring updated knowledge to Long COVID patients.

A previous meta-analysis, including COVID-19 survivors [21], states the superiority of telerehabilitation over no treatment or usual care for dyspnea, limb muscle strength, ambulation capacity, and depression. No significant difference was found in anxiety or quality of life, and no dates were included about pulmonary capacity. No severe adverse events were reported in any of the included studies in this meta-analysis. The authors stated moderate to very low-quality evidence [21]. In our review, we excluded studies where the comparison group received no usual care, no treatment or no rehabilitation (waiting list).

Only randomized clinical trials were included. The methodological analysis of the selected articles showed one study considered “excellent”, three studies considered “good”, one study considered “fair” and one study considered “poor”.

With regard to the adherence of the intervention in this review, it was reported as satisfactory in four articles [37,38,40,42]. In one article [38], program adherence was monitored by a registry, including emails and telephone calls. In one article [40], it was observed that remote supervision could have the potential to improve access to rehabilitation programs, which could increase the motivation of participants. In two articles [39,41], adherence was not evaluated.

On the other side, the absence of adverse effects is common to all included studies and confirms, once again, the positive effects of digital practices on different pathologies already mentioned in previous reviews.

Concerning possible gender differences, which have been previously addressed in some studies [57], the only issue to be highlighted in our findings is that one of the studies showed that females were more fatigued than males and therefore benefited more from the digital physiotherapy practice [41].

Finally, we would like to highlight the use of remote assessment by a telephone call in one article [37]. Furthermore, outcome measures were collected using a self-completed online form in one article [38]. Online assessment may become relevant and effective, as has been stated in previous research during COVID-19 pandemic restrictions [58], but in our results, no other study used digital technology to assess patients.

Notwithstanding, the results of our systematic review should be interpreted in the context of its unique context and PICO criteria, showing differences from previous reviews, as has been highlighted.

### Limitations

Important limitations have been identified in this review. First, limitations due to inclusion criteria mean that only Long COVID adult patients were selected; therefore, under-18-year-old participants are not subject to the review.

Five articles were considered to be of sufficiently high methodological quality; in addition, a low risk of bias was found in four of them. Given the nature of the intervention, only one article [40] used double blinding; therefore, excellent internal validity was limited. The inclusion of very low methodological quality and a high risk of bias in one study [41] limits the overall findings of this review.

The limited scope of the published trials in patients with Long COVID may have led to the conclusion that some studies were simply underpowered to detect a clinically relevant difference. Also, the limited information available on effect size is an important limitation to consider. Moreover, the information regarding dropout rates is limited and should be addressed in depth [39].

## 5. Conclusions

The results of the present review showed that digital physiotherapy practices could be a real opportunity to improve clinical outcomes, including pulmonary, functional, and cardiovascular capacities, quality of life, dyspnea and fatigue in Long COVID patients.

In line with previous findings in various conditions [59], the effectiveness of digital physiotherapy interventions has been proven, and their results have been shown to be no less inferior to those achieved with usual care for Long COVID patients. Therefore, digital physiotherapy could be an effective alternative to usual care and a viable option for providing a safe way of delivering rehabilitation.

This review provides new evidence that digital physiotherapy practice is an appropriate intervention for Long COVID patients and should be recommended in clinical practice guidelines, despite the inherent limitations of the review.

A robust, comprehensive meta-analysis on the effectiveness of digital physiotherapy practice for patients with Long COVID could be available in the future based on the findings.

## Figures and Tables

**Figure 1 healthcare-11-01970-f001:**
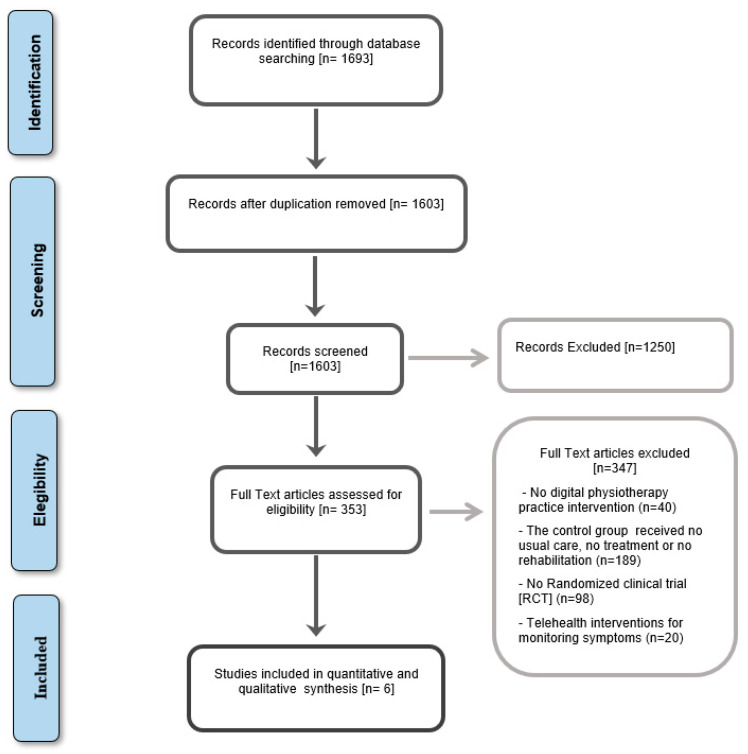
PRISMA Flowchart.

**Figure 2 healthcare-11-01970-f002:**
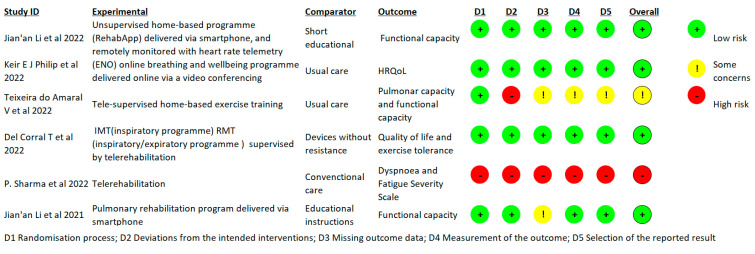
Risk of bias summary [37,38,39,40,41,42].

**Figure 3 healthcare-11-01970-f003:**
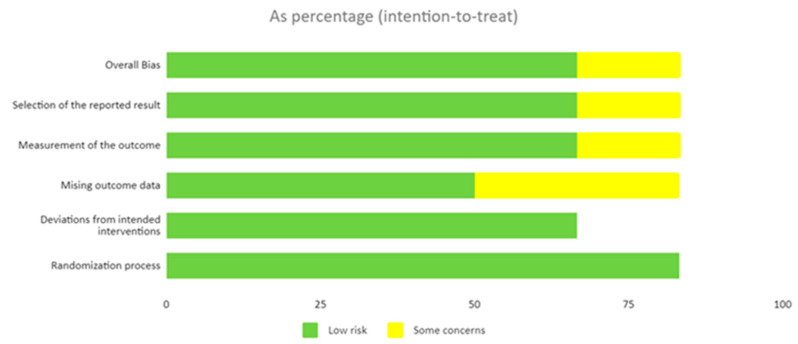
Risk of bias graph.

**Table 1 healthcare-11-01970-t001:** Evaluation of the Methodological Quality of the Selected Studies.

	Jian’an Li et al., 2022 [37]	Keir E J Philip et al., 2022 [38]	Teixeira do Amaral V et al., 2022 [39]	Del Corral T et al., 2022 [40]	P. Sharma et al., 2022 [41]	Jian’an Li et al., 2021 [42]
Eligibility criteria	Y	Y	Y	Y	Y	Y
Randomization	Y	Y	Y	Y	Y	Y
Allocation concealed	Y	Y	N	Y	N	Y
Baseline comparability	Y	Y	Y	Y	N	Y
Subject blinding	N	N	N	Y	N	N
Therapist blinding	N	N	N	Y	N	N
Evaluator blinding	Y	Y	Y	Y	Y	Y
Adequate follow-up	Y	Y	N	Y	N	N
Intention to treat	Y	Y	N	Y	N	Y
Comparison between groups	Y	Y	Y	Y	N	Y
Point estimates and variability	Y	Y	Y	Y	N	Y
Total PeDro Score	8	8	5	9	2	7

PEDro: Physiotherapy Evidence Database. The eligibility criteria do not contribute to the total score. Y: Yes; N: No.

**Table 2 healthcare-11-01970-t002:** Characteristics of Studies.

Author [Year]	Population	Participants [n], Type of Evidence	PeDro Score, Risk of Bias	Intervention	Outcome Measure	Intervention Duration [Weeks]	Follow-Up [Month]	Results
Jian’an Li et al., 2022 [37]	Long COVID Patients	n = 120 [IG 59/CG 61] RCT	8/10—Low	IG: Unsupervised home-based program via smartphone, and monitored with heart rate telemetry CG: Short educational instructions at baseline	Primary outcome: functional exercise capacity by 6 MWT in meters. Secondary outcomes: functional capacity in MMII by squat time 6-MWD in seconds; Pulmonary capacity by spirometry in liters; HRQOL by SF-12 Questionary and dyspnea by mMRC-dyspnea analysis	6	28	TERECO program was superior to control group with regard to functional exercise capacity, LMS and physical HRQOL. The effects could be maintained for a period of 7 months. No differentiation was found in pulmonary function. Improvements were found in the physical component of the SF-12 scale, with effects at post-treatment and follow-up.
Keir E J Philip et al., 2022 [38]	Long COVID Patients	n = 150 [IG 74/CG 76] RCT	8/10—Low	IG: (ENO) online breathing and well-being program online via a video conferencing application CG: Usual care	Primary outcome: HRQOL, by RAND 36-item short form survey instrument mental health composite (MHC) and physical health composite (PHC) scores. Secondary outcome: Pulmonary capacity, visual analog scales (COPD Assessment Test) for breathlessness, and scores on the dyspnea-12, anxiety disorder 7-item scale, and the short form-6D. A thematic analysis exploring participant experience was also conducted using qualitative data from focus groups, survey responses, and email correspondence.	6	--	Improvements in the MHC of quality of life were observed compared to usual care. VAS for breathlessness (running) favored ENO Breathe participation. In the secondary outcomes, no statistically significant differences were observed between the groups. Thematic analysis of participants’ perceptions of the intervention identified three key themes: (1) symptom improvement; (2) the sense that the program complemented standard care; and (3) the particular suitability of singing and music to address their needs. Mind, body, and music-focused practices could influence participants’ recovery.
Teixeira do Amaral V et al., 2022 [39]	Long COVID Patients	n = 32 [IG 12/CG 20] RCT	5/10—Some Concerns	IG: Tele-supervised home-based exercise training	Primary outcome: Anthropometric, Hemodynamic (brachial and central blood pressure) [HR], (arterial stiffness) [BP], vascular capacity (pulse wave velocity) Pulmonary capacity by [PWV] spirometry and functional capacity by (handgrip strength, five-time sit-to-stand [FTSTS], timed up-and-go test [TUG] and six-minute walking test [6 MWT])	12	--	Both the intervention and control groups increased (*p* < 0.001) forced vital capacity (absolute and % of predicted), forced expiratory volume in the first second (absolute and % of predicted) and hand grip strength during follow-up. However, only the intervention group reduced carotid–femoral pulse wave velocity and increased (*p* < 0.05) resting oxygen saturation, mean inspiratory pressure, mean expiratory pressure and % of predicted mean expiratory pressure during follow-up. No significant changes were observed in any other variable during follow-up.
				CG: Usual care				
Del Corral T et al., 2022 [40]	Long COVID Patients	n = 88 [IG (IMT: 22; RMT 22)/CG: IMTsham n = 22, RMTsham n = 22]. RCT	9/10—Low	IG: home-based respiratory muscle training program IMT (inspiratory program) RMT (inspiratory/expiratory program) supervised by telerehabilitation	Primary outcomes: quality of life (EuroQol-5D questionnaire) and exercise tolerance (Ruffier test). Secondary outcomes: respiratory muscle function, physical function 1 min sit-to-stand and dynamometer, pulmonary function, and cognitive and psychological status anxiety/depression levels and post-traumatic stress disorder	8	--	Statistically significant improvement in quality of life, but not in exercise tolerance, in the two training groups compared to the sham groups. The two training groups developed a large statistically significant increase in inspiratory muscle strength and endurance and lower extremity muscle strength compared to the two sham groups. Expiratory muscle strength and peak expiratory flow showed a large, statistically significant increase in the training group.
				CG: Devices without resistance (0 cm H20) lacked threshold valves.				
P. Sharma et al., 2022 [41]	Long COVID Patients	n = 30 [IG: 15/CG:15]. RCT	2/10—High	IG: Telerehabilitation therapeutic protocol	Primary outcomes: Modified Borg Dyspnea Rating Scale and Fatigue Severity Scale	6	--	Results showed that there was a statistically significant difference between IG (MBDS) and CG (MBDS) (*p* = 0.005605 and *p* = 0.01121) and statistically significant difference was found between IG (VAS-F) and CG (VAS-F) (*p* = 0.01818 and *p* = 0.036359).
				CG: Usual care				
Jian’an Li et al., 2021 [42]	Long COVID Patients	n = 120 [IG 59/CG: 60] RCT	7/10—Low	IG: home-based pulmonary rehabilitation program delivered via smartphone. Exercise types comprised breathing control and thoracic expansion, aerobic exercise, and LMS exercise. CG: Educational instructions.	Primary outcome: Functional exercise capacity by 6-MWD in meters. Secondary outcomes: functional capacity by squat time in seconds; pulmonary function by spirometry with parameters being forced expiratory volume in 1 s (FEV1), forced vital capacity (FVC), FEV1/FVC, maximum voluntary ventilation (MVV), and peak expiratory flow; HRQOL measured with SF-12 physical component score (PCS) and mental component score (MCS); and dyspnea by mMRC-dyspnea, favorable outcome (no dyspnea).	6	28	The adjusted between-group difference in change in 6-MWD from baseline was 65.45 m at post-treatment and 68.62 m at follow-up. Treatment effects for LMS were 20.12 s post-treatment and 22.23 s at follow-up. No group differences were found for lung function apart from post-treatment MVV. Increase in SF-12 PCS was greater in the TERECO group, with treatment effects estimated at 3.79 at post-treatment and 2.69 at follow-up. No significant between-group differences were found for improvements in SF-12 in mental component. At post-treatment, 90.4% endorsed a favorable outcome for mMRC dyspnea in the TERECO group vs. 61.7% in control.

IG: Intervention Group; CG: Control Group; 6 MWT: The six-minute walk test; MMII: Lower limbs; 6-MWD: 6-minute walking distance; HRQOL: Health-related quality of life. mMRC-dyspnea: Modified Medical Research Council dyspnea scale; LMS: Lower limb muscle strength; COPD: Chronic obstructive pulmonary disease; Dyspnea-12: Dyspnea-12 questionnaire; The short form-6D: The Short Form 6 Dimension; HR: Heart rate; BP: Blood pressure; PWV: Pulse wave velocity; IMT. Inspiratory muscle training; RMT: Inspiratory and expiratory muscle training; IMT/RMTsham: Inspiratory and respiratory muscle training without resistance; MBDS: Modified Borg Dyspnea Scale; VAS-F: Visual Analogue Scale to Evaluate Fatigue; FEV1: Forced expiratory volume in 1 s; FVC: Forced vital capacity; MVV: Maximum voluntary ventilation; PCS: Physical component score; MCS: Mental component score.

**Table 3 healthcare-11-01970-t003:** Subgroup Analysis by Intervention.

Intervention	Authors and References	Number of Articles	Participants (n),	Low Risk of Bias (% of Total Articles by Intervention)	Risk of Bias: Some Concerns (% of Total Articles by Intervention)	High Risk of Bias (% of Total Articles by Intervention)
Tele-supervised home-based exercise training	Jian’an Li et al., 2022 [37]/Keir E J Philip et al., 2022 [38]/Teixeira do Amaral V et al., 2022 [39]/Del Corral T et al., 2022 [40]	4	233	3 (75%)	1 (25%)	-
Unsupervised home-based program	Jian’an Li et al., 2021 [42]/P. Sharma et al., 2022 [41]	2	74	1 (50%)	-	1 (50%)
Short educationalinstructions	Jian’an Li et al., 2022 [37]/Jian’an Li et al., 2021 [42]	2	121	2 (100%)	-	-
Usual care	Keir E J Philip et al., 2022 [38]/Teixeira do Amaral V et al., 2022 [39]/P. Sharma et al., 2022 [41]	3	111	1 (33.33%)	1 (33.33%)	1 (33.33%)

**Table 4 healthcare-11-01970-t004:** Subgroup Analysis by Outcomes.

Outcomes	Authors and References	Number of Articles	Participants (n)	Low Risk of Bias (% of Total Articles by Outcome)	Risk of Bias: Some Concerns (% of Total Articles by Outcome)	High Risk of Bias (% of Total Articles by Outcome)
Pulmonary capacity	Jian’an Li et al., 2022 [37]/Keir E J Philip et al., 2022 [38]/Teixeira do Amaral V et al., 2022 [39]/Del Corral T et al., 2022 [40]/Jian’an Li et al., 2021 [42]	5	510	4 (80%)	1 (20%)	-
HRQoL (Quality of Life)	Jian’an Li et al., 2022 [37]/Keir E J Philip et al., 2022 [38]/Del Corral T et al., 2022 [40]/Jian’an Li et al., 2021 [42]	4	478	4 (100%)	-	-
Dyspnea	Jian’an Li et al., 2022 [37]/Keir E J Philip et al., 2022 [38]/P. Sharma et al., 2022 [41]/Jian’an Li et al., 2021 [42]	4	420	3 (75%)	-	1 (25%)
Functional capacity	Jian’an Li et al., 2022 [37]/Teixeira do Amaral V et al., 2022 [39]/Del Corral T et al., 2022 [40]/Jian’an Li et al., 2021 [42]	4	358	3 (75%)	1 (25%)	-
Cognitive and psychological status	Keir E J Philip et al., 2022 [38]/Del Corral T et al., 2022 [40]	2	238	2 (100%)	-	-
Fatigue	P. Sharma et al., 2022 [41]	1	30	-	-	1 (100%)
Exercise Tolerance	Del Corral T et al., 2022 [40]	1	88	1 (100%)	-	-
Participants’ experiences	Keir E J Philip et al., 2022 [38]	1	150	1 (100%)	-	-

**Table 5 healthcare-11-01970-t005:** Subgroup Analysis by Results.

	Jian’an Li et al., 2022 [37]	Keir E J Philip et al., 2022 [38]	Teixeira do Amaral V et al., 2022 [39]	Del Corral T et al., 2022 [40]	P. Sharma et al., 2022 [41]	Jian’an Li et al., 2021 [42]
Pulmonary capacity	(+)	(++)	(++)	(++)		(+)
HRQoL (Quality of Life) P (Physical component) M (Mental component)	(++P) (+M)	(+P) (++M)		(++)		(++P) (+M)
Dyspnea	(+)	(+)			(++)	(+)
Functional capacity	(++)		(+)	(++)		(++)
Cognitive and psychological status		(+)		(+)		
Fatigue					(++)	
Exercise Tolerance				(+)		
Cardiovascular function			(++)			
Participants’ experiences		(+)				

(++) Significant difference with Digital Physiotherapy Practice in favor of the experimental versus control group, *p* < 0.05; (+) Positive effect with Digital Physiotherapy Practice (-) No improvement.

## Data Availability

All data relevant to this study are included in the article.

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
