# Peer review of "Effectiveness of Digital Physiotherapy Practice Compared to Usual Care in Long COVID Patients: A Systematic Review"

_healthcare, 2023, doi:10.3390/healthcare11131970_

Round 1
Reviewer 1 Report
Thank you for the opportunity to review the manuscript “Effectiveness of Digital Physiotherapy Practice Compared to Usual Care in Long Covid Patients. A Systematic Review”.
This systematic review included 6 RCTs of moderate to poor methodological quality.
The theoretical assumptions, presented in the introduction, are relevant to explain the study rational.
The Materials and Methods are very well founded. The results present the necessary information for an analysis of each of the included studies (general characterization table). The Subgroup Analysis by Intervention tables also reveal relevant insight that explores different views of the authors and may be of great interest to the reader.
As there is no comment that I consider obligatory, in order to correct any error or to improve any part of the article, I only leave some questions to the authors for reflection and, eventually, to add to the discussion and limitations of the study:
Page 13
Lines 82-83
You are correct! However, you could take the opportunity to discuss the interest of the scientific studies - to serve the health professional (and other professionals) or the people/society/patients/etc.?
Is it more important to understand which modality is more effective or whether the combination of those evidence-based modalities is better for the person/patient? Just a thought...
Lines 101-104
This idea deserved further reflection by the authors because of its relevance and because it could add knowledge and stimulate a discussion within the scientific and clinical community. What do the authors think about this issue?
Lines 112-114
These studies (2016 and 2020) were before Covid-19 pandemic. Is it true now? Please considere this as a "authors backstage discussion" and review the sentence.
Lines 121-126
I'm not sure if it should be here in the "Limitations" that this issue should arise, but:
- what do the authors think about a subject that is currently much questioned and that relates to the evaluation of results and the use of reliable and valid instruments to be used remotely. How, in each of these 6 studies, is the initial, final and follow-up evaluation process described? It could enrich this review if this issue came up in the discussion or even in the limitations.

Author Response
The authors want to thank your thoughtful and constructive comments. All the suggestions were considered and were included into the revised manuscript. Our manuscript has been substantially improved because of the modifications. An itemized point-by-point response to the reviewer’ comments is presented below in attached document

Reviewer 2 Report
The authors state that they conducted a systematic literature review aimed at evaluating the effectiveness of digital physiotherapy practice compared to usual care in patients with Long Covid.
It is noted that the title is quite broad and does not correlate precisely with the PICO question posed. Additionally, the abstract is not structured according to the journal's guidelines.
The extensive use of databases in the review is noteworthy; however, it is suggested to include the search algorithms used for each database. Similarly, in the PRISMA diagram, the reasons for excluding the 353 articles should be indicated.
Regarding the PICO question, it is important to ensure how patients with specific ICD-10 codes were selectively included. Furthermore, the outcomes are not defined according to the guidelines outlined by the Cochrane review.
Table 1 in the Results section is not easily comprehensible. As for Table 2, it merely repeats what is stated in each article without providing a critical analysis of the results.
In the conclusions, it is asserted that remote interventions are equally effective as in-person interventions. However, the evidence presented is inconclusive to support this claim.
The discussion should be improved by providing a more thorough analysis of the results and addressing the study's limitations in greater detail.
Author Response

(The authors gave the same response as above.)

Reviewer 3 Report
In line 147, you said "The review was conducted according to the registered protocol: CRD42022379004" Where was it registered?
No article looked at social differences as a determinant?
Do you think that the use of mobile applications could also help in adherence to digital physiotherapy?
Author Response

(The authors gave the same response as above.)

Reviewer 4 Report
The authors carried out a review regarding digital physiotherapy practice in Long COVID patients in twelve databases until December 2022. The paper is very complete and well writen, with some minor errors found and listed in sequence.
"Effectiveness of Digital Physiotherapy Practice Compared to Usual Care in Long Covid Patients. A Systematic Review" -> "Effectiveness of Digital Physiotherapy Practice Compared to Usual Care in Long Covid Patients: A Systematic Review" (please avoid using "." in the title of the paper)
General comments and minor errors found are listed as follows.
" groups was found." -> " groups were found."
"Digital physiotherapy practice 1; Telerehabilitation 2; Telemedicine 3; Long COVID 4; Persistent COVID syndrome 5" -> "Digital physiotherapy practice; Telerehabilitation; Telemedicine; Long COVID; Persistent COVID syndrome"
"and mood among others" -> "and mood, among others"
"COVID has" -> "COVID has"
"recommended a multidisciplinary rehabilitation approach with strong recommendation" -> please rewrite
"practice, is defined" -> "practice is defined"
"Appendix A. Figure A1" -> "Appendix A, Figure A1"
"Appendix A. Figure A2" -> "Appendix A, Figure A2"
" from 2019" -> please specify the range (from 2019 to ?) (this information is only available in the abstract)
"English, or Spanish" -> "English or Spanish,"
"The intervention must have been an intervention " -> please rewrite
"Comparison: Digital" -> "Comparison: Digital"
"Digital physiotherapy practice compared with usual care and educational care for Long COVID symptoms." -> this sentence seems loose or repeated in the text
"Secondary outcomes: Secondary outcomes may" -> please rewrite
"trials [RCT], were" -> "trials [RCT] were"
"Risk of Bias." -> "Risk of Bias"
"in a Flowchart," -> "in a Flowchart,"
"are presented in Table 1." -> "is presented in Table 1."
"et al" -> "et al."
"et al" -> "et al."
"et al" -> "et al."
"et al" -> "et al."
"et al" -> "et al."
"et al" -> "et al."
please improve quality of Figure 2 (text is blurred)
please improve quality of Figure 3 (text is blurred)
"of included trials." -> "of included trials"
"estimated at" -> "estimated at"
"pressure and" -> "pressure and"
"pressure during" -> "pressure during"
" found between" -> " found between"
"were 20.12 seconds post-treatment " -> "were 20.12 seconds post-treatment "
" seconds at" -> " seconds at"
"include four" -> "included four"
please use "." instead of "," for separating decimals in table 3
" capacity was found " -> " capacity were found "
"3 of them [35,36,38] " -> "3 of them [35,36,38]. "
"From a cognitive and psychological status [35,37], were numerically better in the intervention group than the control, although differences were not statistically significant." -> please rewrite
" in Table 5." -> " in table 5."
" et al " -> " et al. "
" et al " -> " et al. "
" et al " -> " et al. "
" et al " -> " et al. "
" et al " -> " et al. "
" et al " -> " et al. "
"(Quality of Live) (P" -> ?
"(++" -> ?
section 3.3 is not necessary in the paper. please remove intervention
" terms, have" -> " terms have"
" once again the" -> " once again, the"
"[51] the only" -> "[51], the only"
"Is an effective" -> "It is an effective"
Author Response

(The authors gave the same response as above.)

Round 2
Reviewer 2 Report
The authors have provided a revised version of the article that, in my initial evaluation, I had considered unsuitable for publication. I appreciate the efforts made by the authors to address some of the concerns raised, however, there are still notable deficiencies in the text that need to be addressed.
Firstly, it is essential to clearly define the concept of "Long COVID" for the purpose of this review. As recognized by the World Health Organization (WHO), "Long COVID" encompasses a broad spectrum of symptoms and manifestations, including up to 300 different reported symptoms. Therefore, it is crucial to provide a precise definition and criteria to guide the review process and ensure consistency in the evaluation of relevant studies.
Additionally, further clarification is needed regarding the peer review process and the exclusion of 347 articles. It would be beneficial to clearly state the criteria and reasons for excluding these articles in the PRISMA flowchart. This transparency is essential for readers to understand the selection process and evaluate the reliability of the study.
Moreover, the approach to assessing heterogeneity should be elaborated upon. While the authors mention evaluating heterogeneity, they do not provide details on the methods or statistical tests used. It is important to present the relevant statistical analysis, such as forest plots, if heterogeneity is assessed. This information would enhance the transparency and validity of the review findings.
In Table 4, there appears to be a discrepancy between the stated objective of presenting outcomes and the actual presentation of bias assessment results. It is recommended to clearly align the content of the table with the stated objective to avoid confusion for readers.
Furthermore, in Table 5, the assignment of "++" to each article is not described in the Methods section. It would be beneficial to include an explanation of how the ratings were assigned to ensure transparency and reproducibility.
In the conclusion section, while the authors state that digital intervention is effective, the statement is not adequately supported by the reviewed studies or the analysis conducted. It is crucial to provide a balanced and evidence-based conclusion that accurately reflects the findings of the review.
Lastly, the manuscript would greatly benefit from a thorough language revision by a native English speaker with expertise in health-related topics. This would help to improve the overall clarity and readability of the text.
Considering the aforementioned concerns, I would recommend that the authors consider presenting their work not as a systematic review but as a narrative review. This approach would allow for a more comprehensive discussion of the topic while addressing the limitations identified during the review process.
See lines 181, 156, 142, 118, 56
Author Response
Please see the attachment.
Response to Reviewers (Healthcare-2427177)
Dear Reviewer,
The authors want to thank your thoughtful and constructive comments. All the suggestions were considered and were included into the revised manuscript. Our manuscript has been substantially improved because of the modifications. An itemized point-by-point response to the reviewer’ comments is presented below.
RV2: Reviewer
AA: Authors
RV2: The authors have provided a revised version of the article that, in my initial evaluation, I had considered unsuitable for publication. I appreciate the efforts made by the authors to address some of the concerns raised, however, there are still notable deficiencies in the text that need to be addressed. |
RV2: 1. Firstly, it is essential to clearly define the concept of "Long COVID" for the purpose of this review. As recognized by the World Health Organization (WHO), "Long COVID" encompasses a broad spectrum of symptoms and manifestations, including up to 300 different reported symptoms. Therefore, it is crucial to provide a precise definition and criteria to guide the review process and ensure consistency in the evaluation of relevant studies. |
AA: Thanks for your suggestion. Following the reviewer’s recommendation, we have revised the current definitions in depth, update the paragraph and included new paragraphs to add clarity to your requirement: P.1. L.33-37 The syndrome known as Long COVID or persistent COVID has been defined by the World Health Organization (WHO) as a condition that occurs in people with probable or confirmed SARS-CoV-2 infection, with symptoms at least 2 months and that cannot be explained by an alternative diagnosis. Symptoms are wide-ranging and fluctuating 3,4 and can include fatigue and shortness of breath dysfunction over 200 different reported symptoms present against daily functions, job position, health perception, and mood, among others. P.1. L 39-52 The National Health Service in England (NHS) defined Long COVID as the signs and symptoms that develop during or after COVID-19 and continue for more than 12 weeks and are not explained by an alternative diagnosis. To add more information, we include the definition of The National Research Action Plan on Long COVID and the Services and Supports for the Longer-term Impacts of COVID-19 from the United States government who defined Long COVID as signs, symptoms, and conditions that continue or develop after initial COVID-19 or SARS-CoV-2 infection. The signs, symptoms, and conditions are present four weeks or more after the initial phase of infection; may be multisystemic; and may present with a relapsing–remitting pattern and progression or worsening over time, with the possibility of severe and life-threatening events even months or years after infection. It represents many potentially overlapping entities, likely with different biological causes and different sets of risk factors and outcomes. |
RV: 2. Additionally, further clarification is needed regarding the peer review process and the exclusion of 347 articles. It would be beneficial to clearly state the criteria and reasons for excluding these articles in the PRISMA flowchart. This transparency is essential for readers to understand the selection process and evaluate the reliability of the study |
AA: Thanks for your suggestion. Following the reviewer’s recommendation, the PRISMA flowchart has been updated. |
RV2: 3. Moreover, the approach to assessing heterogeneity should be elaborated upon. While the authors mention evaluating heterogeneity, they do not provide details on the methods or statistical tests used. It is important to present the relevant statistical analysis, such as forest plots, if heterogeneity is assessed. This information would enhance the transparency and validity of the review findings. |
AA: Thanks for your suggestion. Following the reviewer’s recommendation, the paragraph was revised, and a new paragraph was included to add clarity to your requirement:
P.7. L.223-234 Articles included in the review shown great heterogeneity in terms of interventions, effect size reported, clinical outcomes and instrument used. As examples, the studies that evaluated lung capacity reveal differences interventions approach as well as in the measurement instruments used. Heterogeneity occurs when there are methodological discordances among the trials included in the review: when the patient populations and the disease are not exactly the same in their characteristics; when the outcome variables used are not defined exactly the same, nor are they measured in exactly the same way; when the interventions applied to the patients are not exactly the same although they bear the same name; and when some of the included trials present some bias in their results as has been previously stated (https://www.elsevier.es/es-revista-revista-senologia-patologia-mamaria--131-avance-resumen-metaanalisis-una-forma-basica-entender-S0214158220300700.
The authors considered that the variability in the patient symptoms, interventions, instruments and outcome measures of the included studies did not allow a sensitive analysis of heterogeneity through a forest plot. Also note that numerous publications with a systematic review design have been published in the Healthcare Journal not including Meta-analysis because of similar lack of heterogeneity (https://www.mdpi.com/search?journal=healthcare&article_type=systematic_review https://www.mdpi.com/2227-9032/11/12/1708 https://www.mdpi.com/2227-9032/11/8/1091
|
RV2: 4. In Table 4, there appears to be a discrepancy between the stated objective of presenting outcomes and the actual presentation of bias assessment results. It is recommended to clearly align the content of the table with the stated objective to avoid confusion for readers. |
AA: Thanks for your suggestion. The Table 4 have been updated |
RV2: 5. Furthermore, in Table 5, the assignment of "++" to each article is not described in the Methods section. It would be beneficial to include an explanation of how the ratings were assigned to ensure transparency and reproducibility.
|
AA: Thanks for your suggestion. The Table 5 has been updated. The authors have included an explanation of how the ratings of the significance were assigned to ensure transparency in Page 12 Lines 42-43: “The level of significance was set at P < 0.05”
|
RV2: 6.- In the conclusion section, while the authors state that digital intervention is effective, the statement is not adequately supported by the reviewed studies or the analysis conducted. It is crucial to provide a balanced and evidence-based conclusion that accurately reflects the findings of the review.
|
AA: Thanks for your suggestion. The authors have reviewed the conclusion in depth. We have no objection to rewrite the sentence:
“The results of the present review seem to indicate that digital physiotherapy practices could be a real opportunity to improve clinical outcomes, including pulmonary, functional, and cardiovascular capacities, quality of life, dyspnoea and fatigue in Long COVID patients based. It could be an effective alternative to usual care and a viable option providing a safe way of delivering rehabilitation”.
|
RV2: 7.- Lastly, the manuscript would greatly benefit from a thorough language revision by a native English speaker with expertise in health-related topics. This would help to improve the overall clarity and readability of the text. |
AA: Thanks for your suggestion. A language revision by a native English speaker physiotherapy and health-related topic has been carried out again through the manuscript.
|
RV2: 8.- Considering the aforementioned concerns, I would recommend that the authors consider presenting their work not as a systematic review but as a narrative review. This approach would allow for a more comprehensive discussion of the topic while addressing the limitations identified during the review process. |
AA: Thanks for your suggestion. Based on Prospero register code, the following of the methodological recommendations of the Cochrane Collaboration Handbook according to the Preferred Reporting Items for Systematic Reviews and Meta-analysis (PRISMA) statement, the review of similar systematic reviews papers published in Healthcare Journal and the comments of the reviewers 1, 3 and 4, the authors are confident with the systematic review design. We do not consider this work to be a narrative review. The authors believe that this manuscript should be considered a systematic review as indicated in the proposed title. None of the other reviewers (1, 3 and 4) have suggested this change in study design. |
RV2: 9.-Comments on the Quality of English Language: See lines 181, 156, 142, 118, 56 |
AA: Thanks for your suggestion. The changes have shown in follow: - A risk of bias graph is show in figure 3 A risk of bias graph is shown in figure 3 - The review was conducted according to the registered protocol: CRD42022379004, although subgroup analyses were planned. The review was conducted according to the registered protocol: CRD42022379004. - In 1 article (38) the adherence to the programme was monitored with a register, including emails and phone calls to individuals who missed sessions In 1 article (38), program adherence was monitored by a registry, including e-mails and telephone calls. - Only randomized clinical trial with control groups were included and methodological analysis shown 1 study considered “excellent'', 3 studies considered “good”, 1 study with considered “fair” and 1 study considered ‘poor. Only randomized clinical trials were included. The methodological analysis of the selected articles showed 1 study considered "excellent", 3 studies considered "good", 1 study considered "fair" and 1 study considered "poor". - , which is an important differentiation from previous reviews. which represents an important differentiation with respect to previous reviews |
